# Assessing the reliability of pediatric emergency medicine billing code assignment for future consideration as a proxy workload measure

Justin M. Park[1,2]*, Erica McDonald[1,2], Yijinmide Buren[2], Gord McInnes[3], Quynh Doan[1,2,4]

1 Faculty of Medicine, University of British Columbia, Vancouver, Canada, 2 British Columbia Children's Hospital Research Institute, Vancouver, Canada, 3 Department of Emergency Medicine, University of British Columbia, Kelowna, Canada, 4 Department of Pediatrics, University of British Columbia, Vancouver, Canada

* jmhpark@student.ubc.ca

## Abstract

**Data Availability Statement:** The data used in this study is sensitive clinical information from real patients. The ethics approval was obtained from the University of British Columbia and BC Women's

### Objectives

Prediction of pediatric emergency department (PED) workload can allow for optimized allocation of resources to improve patient care and reduce physician burnout. A measure of PED workload is thus required, but to date no variable has been consistently used or could be validated against for this purpose. Billing codes, a variable assigned by physicians to reflect the complexity of medical decision making, have the potential to be a proxy measure of PED workload but must be assessed for reliability. In this study, we investigated how reliably billing codes are assigned by PED physicians, and factors that affect the inter-rater reliability of billing code assignment.

### Methods

A retrospective cross-sectional study was completed to determine the reliability of billing code assigned by physicians (n = 150) at a quaternary-level PED between January 2018 and December 2018. Clinical visit information was extracted from health records and presented to a billing auditor, who independently assigned a billing code–considered as the criterion standard. Inter-rater reliability was calculated to assess agreement between the physician-assigned versus billing auditor-assigned billing codes. Unadjusted and adjusted logistic regression models were used to assess the association between covariables of interest and inter-rater reliability.

### Results

Overall, we found substantial inter-rater reliability (AC₂ 0.72 [95% CI 0.64–0.8]) between the billing codes assigned by physicians compared to those assigned by the billing auditor. Adjusted logistic regression models controlling for Pediatric Canadian Triage and Acuity

and Children's Hospital Research Ethics Board, without the intention of sharing the data publicly, even if de-identified. Therefore, the authors are unable to publicly share the data as it will breach the ethics protocol. For data requests, researchers can contact cwreb@bcchr.ubc.ca. As well, all correspondence concerning this manuscript can be addressed to jmhpark@student.ubc.ca.

**Funding:** The authors received no specific funding for this work.

**Competing interests:** The authors have declared that no competing interests exist.

scores, disposition, and time of day suggest that clinical trainee involvement is significantly associated with increased inter-rater reliability.

## Conclusions

Our work identified that there is substantial agreement between PED physician and a billing auditor assigned billing codes, and thus are reliably assigned by PED physicians. This is a crucial step in validating billing codes as a potential proxy measure of pediatric emergency physician workload.

## Introduction

Crowding is a common problem in pediatric emergency departments (PEDs) [1] and can negatively impact patient health outcomes [2–5] and clinicians' wellness [6]. Chan et al. attributes PED crowding in part to inefficiencies in the patient flow—namely, the input, throughput, and output factors [3]. The input and output factors, defined as the number of incoming patients and disposition respectively, are generally not under the control of the PED. However, the rate at which patients are treated, known as the throughput, can be improved by optimizing the allocation of resources such as space and staffing assignments [7, 8]. This can be achieved using a proxy measure to quantify PED physician workload, allowing for prediction of resource needs to guide allocation and ensure efficient PED throughput.

To date, there has been two proposed measures to estimate PED physician workload; however, neither are validated for workload estimation. The first is the time needed to treat, as measured by the direct interaction time spent between the PED physician and the patient [7]. However, workload is determined by a multitude of different factors in addition to the time needed to treat, including mental demand, physical demand, and psychological stress [9–11]. Therefore, time needed to treat by itself cannot adequately represent PED physician workload. Furthermore, it is generally not a conventionally collected variable in the PED and is labour intensive to record, making it largely unavailable for academic and administrative purposes. The second measure, sometimes perceived as a surrogate for physician workload is the Pediatric Canadian Triage and Acuity Score (PaedCTAS). This is a triaging tool that evaluates the urgency of the patient's needs based on their clinical presentation to prioritize access to care in the PED [12]. While the PaedCTAS has been shown to correlate with PED disposition [13], it was not designed to measure workload, nor has it been evaluated for such purposes. Of note, there is evidence to suggest that using the CTAS (adult equivalent of PaedCTAS) alone is not sufficient for determining physician workload in the general emergency department (ED) setting given the large variability in their workload measure at each triage level [14]. This brings into question the validity of using PaedCTAS, a derivative of the original CTAS triaging tool, to be used as a measure of physician workload.

To address the current lack of PED physician workload proxy measure, we propose evaluating billing codes, which are assigned by physicians for compensation either for direct remuneration or shadow estimation of workload and administrative purposes, after each patient encounter based on their impression of the amount of work required to treat the patient. Throughout Canada, many EDs use either a 2 or 3-level billing code system, with greater levels indicating more complexity and work required to manage the patient encounter; some systems also include modifiers which account for other factors such as time of the day, patient age, and procedures performed [15–25]. With the 3-level system, level 1 is assigned for treatment involving a single organ system or a simple condition, level 2 for conditions which necessitate

treatment of at least 2 organ systems with a need for reassessment during the visit, and level 3 for complex conditions requiring prolonged observation and therapy with multiple assessments [15]. In British Columbia (BC) alone, billing code data is used to estimate workload in the fee-for-service setting to allocate approximately $75 million of funding to emergency physicians [26]. Given that billing codes are readily reported and the reliance on billing codes to measure physician workload for remuneration purposes, this variable holds potential to be a proxy measure of PED workload.

To assess if physician assigned billing codes can approximate physician workload, we must evaluate the degree of reliability in which PED physicians are assigning these billing codes. Inter-rater agreement of billing codes has been evaluated in other medical specialties and reliability has been found to vary between them [27–32]. In this study, we aim to assess how reliably PED physicians bill when compared to a billing expert who is also the provincial auditor. In addition, we aim to identify which factors are associated with inter-rater reliability.

## Methods

### Study objective and design

We conducted a retrospective cross-sectional study at BC Children's Hospital (BCCH) ED to evaluate the reliability of billing codes assigned by PED physicians compared to the billing code assigned by a billing auditor, who is one of the listed authors of the research group (G. M.) and does not work at the BCCH. The billing auditor selected is an emergency physician and the Chair for the Fee-For-Service Committee within the Section of Emergency Medicine at Doctors of BC, the association representing physicians in BC. Given the billing auditor's clinical and administrative expertise in emergency medicine, their interpretation of billing code was used as the criterion standard.

The primary objective for this study was to evaluate how consistently billing codes are assigned by determining the inter-rater reliability between PED physician assigned billing codes and billing auditor assigned billing codes. Our secondary objective was to identify visit characteristics associated with inter-rater reliability.

### Study setting and population

BCCH ED is a quaternary care referral centre located in Vancouver, BC with approximately 50,000 annual visits [33]. We collected data from a random sample of visits from children aged up to 18 years who visited the BCCH ED between January 1st, 2018 to December 31st, 2018 inclusive, provided that the patient did not leave without being seen by a physician and that the physician assigned a billing code to their visit. We used health records provided by the Provincial Health Service Authority Data Analytics, Reporting and Evaluation (PHSA DARE) Office. A timeframe before the COVID-19 pandemic was studied to ensure physician billing practices were unaltered by pandemic precautions such as extra PPE and disease screening. The sample of visits was evenly distributed between months of the year and with representation of all 5 levels of the PaedCTAS scale, with propensity for PaedCTAS 3 and 4 as they generally make up the majority of all PED visits [34].

While our PED is staffed with pediatric emergency medicine physicians, general emergency physicians, and nurse practitioners, our study only included visits which were managed or supervised (when a trainee is involved) by a pediatric or general emergency medicine physician, as other care providers do not assign billing codes. Physicians at BCCH ED are paid on an alternate payment plan and therefore utilize the shadow billing system, whereby billing codes are assigned not for remuneration, but for both individual physician performance monitoring and group contract negotiation.

Ethics approval was obtained from the University of British Columbia and BC Women and Children's Hospital Research Ethics Board and the requirement for informed consent was waived by the two ethical governing bodies.

## Outcome measures

The inter-rater reliability between PED physicians and the billing auditor was evaluated using percentage agreement and Gwet's $AC_2$ with 95% confidence intervals (CI) as our primary outcome measure.

As the secondary outcome measure, we calculated the percentage agreement and $AC_2$ values stratified by visit characteristics including triage categories (PaedCATS1-5), patient age (<1y, 1-5y, >5y), whether clinical trainees were involved, time of disposition (day 0800-1800h, evening 1800-2300h, and night 2300-0800h), and disposition (discharged vs. admitted).

## Study procedure

From the chart review, we extracted the billing code assigned by the PED physician and the clinical variables needed for the billing auditor to assign a billing code. Clinical variable selection was informed by consultations with clinicians and published literature around the subject of physician workload. These variables include those which were found to be strong predictors for workload intensity such as the PaedCTAS score, presentations or comorbidities related to mental health, requirement for ambulance, laboratory and imaging ordered, number of subspecialty consultations, procedures performed, need for sedation, trainee involvement, language barrier, disposition, and length of stay [7, 10, 14, 35, 36]. As well, information which can inform the billing auditor of the clinical context were also collected, such as the patient demographic, chief complaint, the history of presenting illness, physical exam findings, vital signs, and any other progress notes or text relevant to the patient visit.

The clinical variables were collected by two trained research students onto the Research Electronic Data Capture platform, a BCCH Research Institute licensed data capture software. The authors had access to patient identifiers, such as the personal healthcare number, during the data extraction which were not collected. To ensure inter-extractor reliability between the students, data extraction training was carried out by a PED physician. Both students separately extracted the data from 15 charts (10% of total sample size) and compared their output for any discrepancies, which were resolved by consensus. This process was repeated until the extracted data between the two students matched for all 15 charts at which point the remaining charts were divided and the data was extracted by each student. In total, 30 charts were extracted in tandem. The data collection was conducted between August to October of 2021.

Following data collection, the billing auditor was given the extracted clinical data to assign a billing code. The billing auditor was blinded to PED physicians' billing codes.

## Analysis approach

We report descriptive statistics to summarize our study sample, using proportions with 95% CI as appropriate. The percentage agreement and Gwet's $AC_2$ statistics were used as the measure of reliability in the PED physicians' billing practices. The $AC_2$ statistic was chosen for its resiliency against the effects of trait prevalence, where high chance agreement can paradoxically result in low chance-corrected agreement despite relatively higher percentage agreement [37, 38]. The Landis and Koch criterion was used to interpret the $AC_2$ values which categorizes the chance-corrected agreement statistics as follows: 0–0.20 slight agreement, 0.21–0.40 fair, 0.41–0.60 moderate, 0.61–0.80 substantial, and >0.80 excellent agreement [39]. $AC_2$ was calculated with linear weighting.

We completed univariate logistic regression models to determine the impact of covariates of interest on inter-rater reliability, then adjusted potential confounders. Analyses were performed using R statistical software.

Given that there are two raters (PED physicians and the billing auditor) and three categories (billing codes 1, 2, 3), to estimate AC2 within a margin of 0.15 with 95% confidence, a sample size of 90 was required [40]. We added a margin to ensure that we obtain 150 chart visits that meet all our eligibility criteria and requested 300 randomly selected charts meeting specified distribution over time and acuity from the PHSA DARE Office. Upon receipt, we used the Microsoft Excel's random number generator function and reviewed charts in a randomized sequence to review for eligibility and extract data until the sample size of 150 was met.

## Results

We requested 300 randomized patient records from the PHSA DARE Office, and reached the sample size requirement of 150 after reviewing 187 charts (Fig 1). The distribution of subgroups across our sample is outlined in Table 1.

Overall, the percent agreement between PED physician and billing auditor was 68.7%. There was substantial inter-rater reliability (AC2: 0.72 95% CI: 0.64, 0.8). Among the 47 (31.3%) instances where the PED physician and the billing auditor disagreed, the PED physician assigned a lower billing code than the billing auditor 27 times (18%).

Table 1 shows the inter-rater reliability indices for the overall sample size and stratified by visit characteristics. The inter-rater reliability is highest in PaedCTAS 3 (AC$_2$: 0.84 95% CI: 0.6,

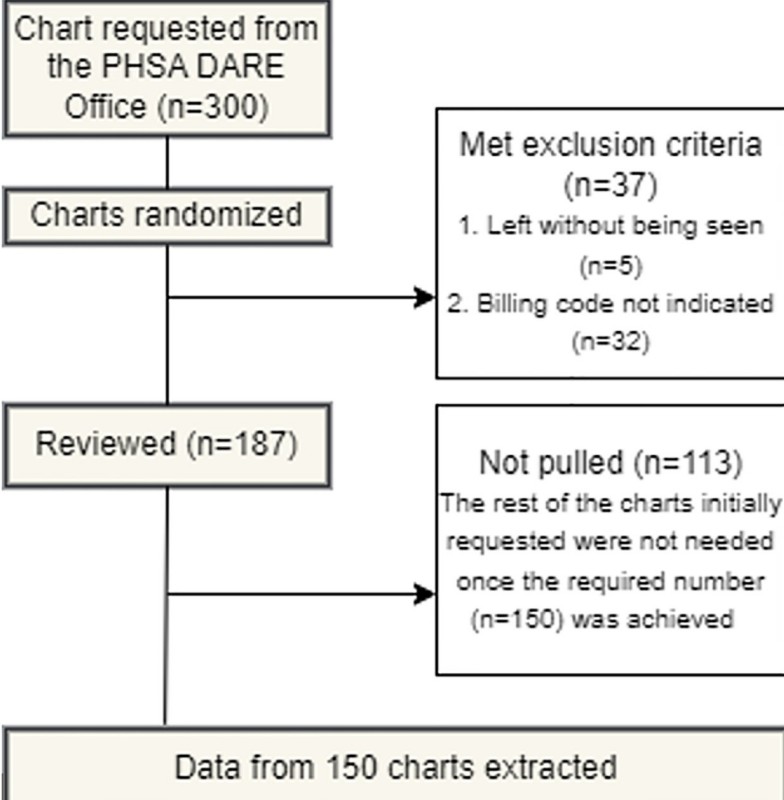

**Fig 1. Details of the chart review process.**

**Table 1. Degree of billing code assignment agreement and reliability stratified by visit characteristics.**

| Analysis Group | n (%) | AC$_2$ (95% CI) | % Agreement |
|---|---|---|---|
| Overall | 150 (100) | 0.72 (0.64, 0.8) | 68.7 |
| PaedCTAS 1 | 19 (12.7) | 0.26 (-0.3, 0.8) | 57.9 |
| PaedCTAS 2 | 33 (22.0) | 0.75 (0.6, 0.9) | 66.7 |
| PaedCTAS 3 | 36 (24.0) | 0.84 (0.7, 0.98) | 80.6 |
| PaedCTAS 4 | 39 (26.0) | 0.34 (0.01, 0.7) | 61.5 |
| PaedCTAS 5 | 23 (15.3) | 0.49 (0.1, 0.9) | 73.9 |
| Age <1 y | 28 (18.7) | 0.81 (0.7, 0.95) | 71.4 |
| Age 1–5 y | 55 (36.7) | 0.68 (0.5, 0.8) | 63.6 |
| Age 6+ y | 67 (44.6) | 0.71 (0.6, 0.8) | 71.6 |
| Trainees involved | 83 (55.3) | 0.79 (0.7, 0.9) | 75.9 |
| Trainees not involved | 67 (44.7) | 0.61 (0.5, 0.7) | 59.7 |
| Day (0800–1800) | 53 (35.3) | 0.7 (0.6, 0.8) | 67.9 |
| Evening (1800–2300) | 48 (32.0) | 0.75 (0.6, 0.9) | 70.8 |
| Night (2300–0800) | 49 (32.7) | 0.71 (0.6, 0.9) | 67.3 |
| Discharged | 130 (86.7) | 0.76 (0.7, 0.8) | 70.7 |
| Admitted | 20 (13.3) | 0.58 (0.3, 0.9) | 55 |

0.9), age <1y (AC$_2$: 0.81 95% CI: 0.7, 0.95), and clinical trainee involvement (AC$_2$: 0.79 95% CI: 0.7, 0.9) subgroups. Other subgroups display wide and overlapping CIs and no pattern of changes in the inter-rater reliability index.

Table 2 shows the adjusted and unadjusted regressions exploring the association between individual visit characteristics and inter-rater reliability. After controlling for all other subgroups in the adjusted model, clinical trainee involvement is the only subgroup showing significant association with increased billing code assignment reliability (adjusted OR: 2.2 95% CI: 1.02, 4.9), when compared to visits managed only by the staff PED physician.

## Discussion

### Interpretation

Our study found substantial inter-rater reliability in billing code assignment between PED physicians and the billing auditor, which suggests billing codes are accurately assigned. This is

**Table 2. Unadjusted and adjusted logistic regression models of association between subgroups and increased inter-rater reliability.**

| Reference Subgroup | Analyzed Subgroup | Unadjusted OR (95% CI)[a] | Adjusted OR (95% CI) |
|---|---|---|---|
| No trainee involvement | Trainee involved | 2.1 (1.1, 4.3) | 2.2 (1.02, 4.9) |
| PaedCTAS 1 | PaedCTAS 2 | 1.5 (0.4, 4.7) | 1.1 (0.3, 4.1) |
| | PaedCTAS 3 | 3.0 (0.9, 10.6) | 2.2 (0.5, 9.7) |
| | PaedCTAS 4 | 1.2 (0.4, 3.6) | 0.9 (0.2, 3.8) |
| | PaedCTAS 5 | 2.1 (0.6, 7.9) | 1.3 (0.2, 6.5) |
| Age < 1y | 1–5 | 0.7 (0.2, 2.0) | 0.8 (0.3, 2.3) |
| | 6+ | 1.3 (0.5, 3.5) | 1.8 (0.6, 5.4) |
| Day | Evening | 1.1 (0.5, 2.7) | 1.4 (0.6, 3.7) |
| | Night | 1.0 (0.4, 2.2) | 1.2 (0.5, 3.2) |
| Admitted | Discharged | 2.0 (0.7, 5.2) | 1.5 (0.4, 5.8) |

[a]. OR: Odd's ratio; CI: Confidence Interval

an important step in establishing the potential for billing codes to serve as a proxy measure of PED workload. While several subgroups showed association with higher inter-rater reliability, only clinical trainee involvement was found to be associated with significantly higher inter-rater reliability, and this significance persisted when controlling for PaedCTAS, patient age, time of day, and disposition.

Several studies evaluating billing practices showed that the amount of experience and time allocated to teaching physicians about billing is associated with increased billing accuracy, such that staff and senior residents tend to have greater levels of comfort and knowledge in assigning billing codes compared to junior residents [26, 28, 39, 40]. These findings are rather intuitive, as more exposure and education regarding a certain topic understandably increases one's competency in practice. Therefore, given that billing codes in BCCH ED are only assigned by staff physicians, our results finding high inter-rater reliability with the billing auditor is expected. A systematic review analyzing current billing practices to recommend methods of improving pediatric billing accuracy supports this notion, stating that more billing education is a key component to improved accuracy [41]. Other studies that evaluate billing practices, which yielded findings of lower billing accuracy, include residents or recent residency graduates to assess their quality of education, readiness, and the financial impact of inaccurate billing, rather than assessing billing reliability by experienced staff [27–31].

Our results also show significantly increased odds of higher inter-rater agreement when clinical trainees are involved, which may be explained by a few factors. First, PED clinical trainees' documentation has been reported to be more complete when compared to staff physicians [42] which may have allowed the billing auditor to have a better context and more accuracy in assigning their billing code, increasing the probability of agreement. Second, clinical trainee education and participation is at times intentionally set aside to prioritize the efficiency of patient flow when the ED capacity is stressed [43, 44]. In these cases, it may be that trainees are more likely be involved in simpler cases which require less interpretation to code. This appears to be reflected in our samples as high acuity cases, which are more likely to be complex, involve fewer trainees than low acuity cases.

We acknowledge that the immense complexity of estimating PED workload cannot be entirely addressed using a 3-level billing system. However, until a more comprehensive PED workload measure is developed and validated, it may be the most appropriate and accessible variable for physician workload estimation given the following reasons. First, billing codes are by design meant to estimate the complexity of clinical decision making and treatment, which is demonstrated in their utility as the variable used to allocate millions of dollars to compensate fee-for-service physicians. Second, the 3-level billing code system is widely used in Canada, as it is implemented in BC, Ontario, Prince Edward Island, and the Northwest Territories [15, 18, 19, 21]. Furthermore, compared to the 2-level system used in other provinces such as Quebec, Newfoundland and Labrador, Saskatchewan, and Manitoba [16, 22, 24, 25], the 3-level system may offer better stratification in estimating the PED workload. Third, a 3-level billing system can be simple to learn and assign in comparison to other existing billing systems, which may be contributing to its high reliability in use by PED physicians. More complex billing systems exist in other specialties which is based off a diverse set of diagnostic or procedural work, such as the International Classification of Diseases or Current Procedural Terminology in the United States, or provincial payment schedules in Canada [45]. These contain thousands of billing codes plus modifiers, which are constantly changing and can often be challenging for physicians to use [46].

## Limitations

Our study results should be interpreted within its limitations. First, the billing auditor's billing code assignment depends on the quality of the physician's documentation. Within our sample, the billing auditor flagged seven of the 150 patient records, indicating that there is poor documentation. In two of the seven flagged records, the alternative billing code they would have assigned, had the documentation contained the required details, matched with the physician's billing code assignment. Therefore, it is possible that improving physician documentation will likely increase the inter-rater reliability, and that our reported level of agreement, based on retrospective documentation, may be conservative.

Secondly, further research with additional sample sizes in various visit characteristics is needed to explore the potential association between them and PED physician billing code reliability. Our study does not assess whether the association found between trainee involvement and improved billing code accuracy is intrinsic to the trainee's charting or if the association can be explained by other variables.

Finally, we used shadow billing codes from PED physicians and their compensation is not dependent on billing pattern. Without a direct financial incentive, concerns may arise about the accuracy of shadow billing data [47]. However, a study in Alberta showed that shadow billing does not affect the accuracy at which they are submitted by specialists including pediatricians in urban, acute care hospitals compared to fee-for-service billing [48]. This suggests that the use of shadow billing data in our study is unlikely to affect the validity of our results.

## Conclusion

In this study, we showed that 3-level billing codes are accurately assigned by PED physicians. This provides a positive first step in the validation of billing codes as a proxy measure of PED workload. With a validated proxy measure, opportunities exist for better optimization of PED resource allocation via workload prediction, which can ultimately improve the throughput.

## Acknowledgments

We thank Jeffrey Bone and Punit Virk for their valuable feedback and assistance with the statistical analysis of our study.

## Author Contributions

**Conceptualization:** Erica McDonald, Quynh Doan.

**Data curation:** Justin M. Park, Yijinmide Buren, Gord McInnes.

**Formal analysis:** Justin M. Park, Erica McDonald, Yijinmide Buren, Quynh Doan.

**Investigation:** Justin M. Park, Quynh Doan.

**Methodology:** Justin M. Park, Erica McDonald, Quynh Doan.

**Project administration:** Justin M. Park, Quynh Doan.

**Supervision:** Quynh Doan.

**Writing – original draft:** Justin M. Park, Erica McDonald, Yijinmide Buren.

**Writing – review & editing:** Justin M. Park, Erica McDonald, Quynh Doan.

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
