## [Decision Letter · Decision Letter 0]

5 May 2023

PONE-D-23-06963Assessment of billing code as a proxy measure for pediatric emergency department workloadPLOS ONE

Dear Dr. Park,

Thank you for submitting your manuscript to PLOS ONE. After careful consideration, we feel that it has merit but does not fully meet PLOS ONE’s publication criteria as it currently stands. Therefore, we invite you to submit a revised version of the manuscript that addresses the points raised during the review process.

Please address all the reviewer comments and submit a revised version within the stipulated time below. 

We look forward to receiving your revised manuscript.

Kind regards,

Benjamin Demah Nuertey, MD, MPH, MA, MWACP, FWACP

Academic Editor

PLOS ONE

Journal Requirements:

Additional Editor Comments:

1. Line 94-96 stated an aim: “In this study, we aim to assess how reliably PED physicians bill when compared to a billing expert and provincial auditor, and identify which factors are associated with inter-rater reliability.” Which is different from that stated elsewhere in the manuscript. This aim suggests three comparisons contrary to what was done.

2. It was stated in lines 111- 115, that, “BCCH ED is a quaternary care referral centre located in Vancouver, BC with approximately 50,000 annual visits (30). We collected data from a random sample of visits from children aged up to 18 years who visited the BCCH ED between January 1st, 2018 to December 31st, 2018 inclusive, provided that the patient did not leave without being seen by a physician and that the physician assigned a billing code to their visit.”

a. More details on how the 150 was selected from the 50000 will be much appreciated to enable your readership to make a judgement on selection bias.

b. From the low chart, you requested 300 charts from the PHSA DARE, please explain more clearly how those 300 charts were selected from an average 50,000.

3. Please provide more information describing how the billing code is assigned to enable your international readers to understand the study being presented.

4. Discussion section, sub section “interpretation”: lines 202- 204 stated that: “Our study found substantial inter-rater reliability in billing code assignment between PED physicians and the billing auditor, which suggests billing codes are accurately assigned, and may reliably serve as a proxy measure of PED workload.”

a. The portion of the interpretation: “may reliably serve as a proxy measure of PED workload” is outside the scope of your study and you cannot make that conclusion.

b. The study accurately determined the interrater reliability of billing code assignment between PED physicians in a tertiary health care facility and an external pediatric emergency physician.

5. If the authors are interested in this current title of “Assessment of billing code as a proxy measure for pediatric emergency department workload” then the methodology should compare billing code and the time needed to treat, as measured by the direct interaction time spent between the PED physician and the patient.

6. You rightly stated in lines 67-68 that “the time needed to treat, as measured by the direct interaction time spent between the PED physician and the patient is generally not a conventionally collected variable in the PED and is labour intensive to record, making it largely unavailable for academic and administrative purposes.”

a. But this direct measure of PED workload is more of a “gold standard” measure for PED workload compared to an audit of a retrospective data on physician billing which is limited by the quality of physician documentation.

b. Secondly, this study compares a real time measure of PED physician billing code with a proxy measure done later by a third party who did not take care of the patient.

c. I would consider the PED physician billing code as the gold standard instead of the external auditor who never saw the patient in question and was working with limited recorded information.

d. I therefore suggest a change of the title to reflect the work that was done.

7. Provide more detail on the clinical variables collected and how they were selected.

8. Check for consistency in language and formatting throughout the section.

9. Is the independent auditor listed as a member of the team of authors of this paper? If so, please state that clearly. If not, then he should be acknowledged in the acknowledgement.

Reviewers' comments:

Please find below the response and comments from the reviewers

**Comments to the Author**

1. Is the manuscript technically sound, and do the data support the conclusions?

Reviewer #1: Yes

Reviewer #2: Yes

2. Has the statistical analysis been performed appropriately and rigorously? 

Reviewer #1: Yes

Reviewer #2: Yes

3. Have the authors made all data underlying the findings in their manuscript fully available?

Reviewer #1: No

Reviewer #2: Yes

4. Is the manuscript presented in an intelligible fashion and written in standard English?

Reviewer #1: Yes

Reviewer #2: Yes

5. Review Comments to the Author

Reviewer #1: General Comments:

Thank you very much for the opportunity to review this paper. Work load in emergency departments is an important factor that affects outcomes for both patients and doctors. Overall, the concept of the study is sound, the study is well designed and the paper well written. The methodology and selection of participants and comparator as well as the statistical analysis are appropriate. The conclusion drawn are also supported by the findings. I, however have a few comments on the paper:

1. The study found that there was good inter-rater reliability in billing codes assigned by doctors and a billing code auditor and so this may be a good proxy to measure physician workload, but it does not discuss how billing code could be employed to measure workload. Granted, determining inter-rater reliability was the primary objective of the study, but a little discussion of how billing codes could translate into measures of workload as a whole will be helpful to readers.

2. In line 83, it is acknowledged that there are different billing code systems, i.e., 2 or 3 level systems. Will the use of either a 2 or 3 level billing system, or any other modifiers in the billing code system have an effect on workload categorization if billing code is used as a proxy for work load. How would this impact uniformity in work load determination across institutions and countries? Could there be uniformity in the determination of workload?

3. In Lines 237, 239 and 241 you mention that the three-level billing system is widely used in Canada and that it offers better stratification in determining PED workload and is easy to teach, so would what you are proposing work in only areas with a 3-level coding system, or do you propose that areas in which different billing systems are used should change to a 3-level system for uniformity?

Reviewer #2: The manuscript is well written with clear objective, result analysis and discussion.

Besides the auditor from another institution, it is advisable to be stated clearly if he/she has no other association with the study.

Could the evaluation by two independent auditors give more reliable results?

References: Many of the referenced articles are beyond a decade. Could the authors cite more recent articles where applicable?

6. PLOS authors have the option to publish the peer review history of their article (what does this mean?). If published, this will include your full peer review and any attached files.

Reviewer #1: No

Reviewer #2: No

---

## [Author Response · Author response to Decision Letter 0]

29 Jun 2023

Dear Dr. Benjamin Demah Nuertey,

Academic Editor

PLOS ONE

Thank you and the reviewers for your insightful comments and the opportunity to revise and improve our manuscript for resubmission. Below you will find our responses and corresponding revisions made in our manuscript. Please note, references to specific lines in this document are referring to the revised manuscript when the track changes are hidden. 

Response: Thank you. This has been revised. 

Response: Our revised cover letter outlines that we are unable to publicly release our data even if de-identified as this was not agreed upon with the research ethics board upon obtaining our ethics approval. To release this data would be a breach of our ethics protocol.

3. Line 94-96 stated an aim: “In this study, we aim to assess how reliably PED physicians bill when compared to a billing expert and provincial auditor, and identify which factors are associated with inter-rater reliability.” Which is different from that stated elsewhere in the manuscript. This aim suggests three comparisons contrary to what was done.

Response: Thank you for identifying this source of potential confusion. We believe our wording of this section may been read as if we were aiming to compare the PED physician billing to a billing expert as well as a provincial auditor. However, our intention was to state that the billing expert is also the acting provincial auditor. We have amended the introduction section to eliminate this potential source of confusion, which now reads:

Lines 97-99 “In this study, we aim to assess how reliably PED physicians bill when compared to a billing expert who is also the provincial auditor. In addition, we aim to identify which factors are associated with inter-rater reliability.”

4. It was stated in lines 111- 115, that, “BCCH ED is a quaternary care referral centre located in Vancouver, BC with approximately 50,000 annual visits (30). We collected data from a random sample of visits from children aged up to 18 years who visited the BCCH ED between January 1st, 2018 to December 31st, 2018 inclusive, provided that the patient did not leave without being seen by a physician and that the physician assigned a billing code to their visit.”

a. More details on how the 150 was selected from the 50000 will be much appreciated to enable your readership to make a judgement on selection bias.

b. From the low chart, you requested 300 charts from the PHSA DARE, please explain more clearly how those 300 charts were selected from an average 50,000.

Response: The PHSA DARE office acts as a central data repository which provides clinical data according to the specified date range and sample characteristics requested by researchers. While our sample size estimate indicated the need for 150 entries, 300 charts were requested from PHSA DARE office to allow for an ample margin of safety in anticipation of receiving charts which do not meet our inclusion criteria. PHSA DARE used a random generator to select 300 charts with the characteristics required as detailed in our manuscript in lines 122-125 (…evenly distributed between months of the year and with representation of all 5 levels of the PaedCTAS scale, with propensity for PaedCTAS 3 and 4 as they generally make up the majority of all PED visits (34).”). Upon receipt of the 300 chart visits, we used a random number generator on Excel to review charts in the random sequence to extract data until 150 sample size was fulfilled. 

We have amended our methods sections to further clarify our sample selection process explained above:

Lines 181-187 “Given that there are two raters (PED physicians and the billing auditor) and three categories (billing codes 1, 2, 3), to estimate AC2 within a margin of 0.15 with 95% confidence, a sample size of 90 was required (37). We added a margin to ensure that we obtain 150 chart visits that meet all our eligibility criteria and requested 300 randomly selected charts meeting specified distribution over time and acuity, from the PHSA DARE Office. Upon receipt, we used the Microsoft Excel’s random number generator function and reviewed charts in a randomized sequence to review for eligibility and extract data until the sample size of 150 was met.” 

5. Please provide more information describing how the billing code is assigned to enable your international readers to understand the study being presented.

Response: We have added details specifying the level of complexity represented by each billing category in the introduction section. The amended section now reads:

Lines 82-93 “Throughout Canada, many EDs use either a 2 or 3-level billing code system, with greater levels indicating more complexity and work required to manage the patient encounter; some systems also include modifiers which account for other factors such as time of the day, patient age, and procedures performed (15-25). With the 3-level system, level 1 is assigned for treatment involving a single organ system or a simple condition, level 2 for conditions which necessitate treatment of at least 2 organ systems with a need for reassessment during the visit, and level 3 for complex conditions requiring prolonged observation and therapy with multiple assessments (15). In British Columbia (BC) alone, billing code data is used to estimate workload in the fee-for-service setting to allocate approximately $75 million of funding to emergency physicians (26). Given that billing codes are readily reported and the reliance on billing codes to measure physician workload for remuneration purposes, this variable holds potential to be a proxy measure of PED workload.” 

6. Discussion section, sub section “interpretation”: lines 202- 204 stated that: “Our study found substantial inter-rater reliability in billing code assignment between PED physicians and the billing auditor, which suggests billing codes are accurately assigned, and may reliably serve as a proxy measure of PED workload.”

a. The portion of the interpretation: “may reliably serve as a proxy measure of PED workload” is outside the scope of your study and you cannot make that conclusion.

b. The study accurately determined the interrater reliability of billing code assignment between PED physicians in a tertiary health care facility and an external pediatric emergency physician.

Response: We agree that evidence of high reliability in billing code assignment does not fully validate billing codes as a proxy measure of PED physician workload. For adequate proxy measure validation, additional studies investigating the association between billing codes and clinical variables known to represent workload is required, which is currently underway within our team. To reflect this, we have edited our interpretation section which now reads:

Lines 215-221 “Our study found substantial inter-rater reliability in billing code assignment between PED physicians and the billing auditor, which suggests billing codes are accurately assigned. This is an important step in establishing the potential for billing codes to serve as a proxy measure of PED workload. While several subgroups showed association with higher inter-rater reliability, only clinical trainee involvement was found to be associated with significantly higher inter-rater reliability, and this significance persisted when controlling for PaedCTAS, patient age, time of day, and disposition.”

Other sections which required similar changes include:

Lines 46-48 in the Abstract: “Our work identified that there is substantial agreement between PED physician and a billing auditor assigned billing codes, and thus are reliably assigned by PED physicians. This is a crucial step in validating billing codes as a potential proxy measure of pediatric emergency physician workload.”

7. If the authors are interested in this current title of “Assessment of billing code as a proxy measure for pediatric emergency department workload” then the methodology should compare billing code and the time needed to treat, as measured by the direct interaction time spent between the PED physician and the patient.

Response: We acknowledge that the title should be more specific to the study objective and have changed it to: “Assessing the reliability of pediatric emergency medicine billing code assignment for future consideration as a proxy workload measure”. We have also made clearer in the background that there is no agreed upon gold standard measure for workload and time spent with a patient should not be considered as such. While time it is a component of workload, it does not consider the mental or physical demand, physician performance, effort, or frustration, and is thus an incomplete measure of workload. The background section referring to time spent with patient under the introduction section is now revised to read: 

Lines 61-66 “To date, there has been two proposed measures to estimate PED physician workload; however, neither are validated for workload estimation. The first is the time needed to treat, as measured by the direct interaction time spent between the PED physician and the patient (7). However, workload is determined by a multitude of different factors in addition to the time needed to treat, including mental demand, physical demand, and psychological stress (9–11). Therefore, time needed to treat by itself cannot adequately represent PED physician workload.”

Further as we clarified the objective of this study, which is to establish the inter-rater reliability of billing code assignment by clinicians in the field, the criterion standard of choice is a billing auditor, as used in practice to verify physicians’ billing accuracy. In our study, the auditor used similar practices and approaches as when functioning as an auditor on behalf of the Ministry of Health.

8. You rightly stated in lines 67-68 that “the time needed to treat, as measured by the direct interaction time spent between the PED physician and the patient is generally not a conventionally collected variable in the PED and is labour intensive to record, making it largely unavailable for academic and administrative purposes.”

a. But this direct measure of PED workload is more of a “gold standard” measure for PED workload compared to an audit of a retrospective data on physician billing which is limited by the quality of physician documentation.

Response: As shared in response to comment #5, time spent with patient is not a valid gold standard for workload in general, but also not the criterion standard for the evaluation of inter-rater reliability for billing code assignment.

b. Secondly, this study compares a real time measure of PED physician billing code with a proxy measure done later by a third party who did not take care of the patient. I would consider the PED physician billing code as the gold standard instead of the external auditor who never saw the patient in question and was working with limited recorded information. I therefore suggest a change of the title to reflect the work that was done.

Response: As shared in response to comment #5, the billing auditor regularly verifies physicians’ billing accuracy on behalf of the Ministry of Health and has used a similar approach working with the Ministry to designate a billing code. This was therefore our criterion standard. 

9. Provide more detail on the clinical variables collected and how they were selected.

Response: We have amended the methods section, under study procedure subsection to the following:

Lines 145-155 “From the chart review, we extracted the billing code assigned by the PED physician and the clinical variables needed for the billing auditor to assign a billing code. Clinical variable selection was informed by consultations with clinicians and published literature around the subject of physician workload. These variables include those which were found to be strong predictors for workload intensity such as the PaedCTAS score, presentations or comorbidities related to mental health, requirement for ambulance, laboratory and imaging ordered, number of subspecialty consultations, procedures performed, need for sedation, trainee involvement, language barrier, disposition, and length of stay (7,10,14,35,36). As well, information which can inform the billing auditor of the clinical context were also collected, such as the patient demographic, chief complaint, the history of presenting illness, physical exam findings, vital signs, and any other progress notes or text relevant to the patient visit.” 

10. Check for consistency in language and formatting throughout the section.

Response: Thank you for this feedback. We apologize for any inconsistencies that may have been present in our original submission and have reviewed our manuscript carefully to revise our format according to the PLOS ONE sample template. 

11. Is the independent auditor listed as a member of the team of authors of this paper? If so, please state that clearly. If not, then he should be acknowledged in the acknowledgement.

Response: Yes, the independent auditor is listed as one of the authors for this manuscript. We have amended the methods section to:

Lines 102-105 “We conducted a retrospective cross-sectional study at BC Children’s Hospital (BCCH) ED to evaluate the reliability of billing codes assigned by PED physicians compared to the billing code assigned by a billing auditor, who is one of the listed authors of the research group (G.M.) and does not work at the BCCH.”

Comments from reviewer 1:

1. The study found that there was good inter-rater reliability in billing codes assigned by doctors and a billing code auditor and so this may be a good proxy to measure physician workload, but it does not discuss how billing code could be employed to measure workload. Granted, determining inter-rater reliability was the primary objective of the study, but a little discussion of how billing codes could translate into measures of workload as a whole will be helpful to readers.

For now, this study suggests that billing codes can potentially act as a proxy measure of PED physician workload, however additional studies for further validation are required as discussed in our response to comment 4ab. We refrained from discussing the details of how we foresee billing codes being used as a proxy measure to prevent misleading statements which are not found in our current study. Future work can explore further the association between billing codes and known PED workload predictors. If such a study also yields supportive evidence to suggest billing codes can be a proxy measure of PED physician workload, it may be possible to use billing codes as a proxy measure of PED workload to objectively guide resource allocation, budget negotiation, and workforce recruitment. 

2. In line 83, it is acknowledged that there are different billing code systems, i.e., 2 or 3 level systems. Will the use of either a 2 or 3 level billing system, or any other modifiers in the billing code system have an effect on workload categorization if billing code is used as a proxy for work load. How would this impact uniformity in work load determination across institutions and countries? Could there be uniformity in the determination of workload?

Response: Other billing systems or addition of modifiers are not within the scope of our study. However, the 3-level system is widely used in many Canadian provinces such that its validation will still yield a proxy measure which is generalizable for the numerous PEDs in Canada that utilize this billing system. 

3. In Lines 237, 239 and 241 you mention that the three-level billing system is widely used in Canada and that it offers better stratification in determining PED workload and is easy to teach, so would what you are proposing work in only areas with a 3-level coding system, or do you propose that areas in which different billing systems are used should change to a 3-level system for uniformity?

Response: In Canada, the management of the healthcare system is under the jurisdiction of the provincial government, therefore physician billing and compensation are also handled provincially. We are not suggesting uniformity of billing codes across all provinces but are inferring that physician billing is reliable in a 3-level coding system in the select provinces in which it is used. Although the alternative 2-level billing system was not explicitly studied, it may be reasonable to state that given fewer levels, the 2-level billing system has less variability therefore potentially more inter-rater reliability compared to the 3-level system. 

Comments from reviewer 2:

1. Besides the auditor from another institution, it is advisable to be stated clearly if he/she has no other association with the study.

Response: Please see our response for comment #9 from the Academic Editor. 

2. Could the evaluation by two independent auditors give more reliable results?

Response: Determination of inter-rater reliability is commonly performed by comparing the sample against a single criterion standard to estimate accuracy. Therefore, to achieve the objective of our study, interpretation from a single auditor was sufficient.

3. References: Many of the referenced articles are beyond a decade. Could the authors cite more recent articles where applicable?

Response: We have reviewed our references and have added more recent references to strengthen our paper where applicable: 

1. Added to line 53:

Jang H, Ozkaynak M, Ayer T, Sills MR. Factors Associated with First Medication Time for Children Treated in the Emergency Department for Asthma. Pediatr Emerg Care. 2021 Jan 1;37(1):E42–7. 

2. Added to line 58: 

Vanbrabant L, Braekers K, Ramaekers K. Improving emergency department performance by revising the patient–physician assignment process. Flex Serv Manuf J. 2021 Sep 1;33(3):783–845. 

3. Added to line 65:

Fishbein D, Nambiar S, Mckenzie K, Mayorga M, Fitts EP, Miller K, et al. Objective measures of workload in healthcare: a narrative review. Int J Health Care Qual Assur. 2020;33(1):1–17.

---

## [Decision Letter · Decision Letter 1]

15 Aug 2023

Assessing the reliability of pediatric emergency medicine billing code assignment for future consideration as a proxy workload measure

PONE-D-23-06963R1

Dear Dr. Park,

We’re pleased to inform you that your manuscript has been judged scientifically suitable for publication and will be formally accepted for publication once it meets all outstanding technical requirements.

Kind regards,

Benjamin Demah Nuertey, MD, MPH, MA, MWACP, FWACP

Academic Editor

PLOS ONE

Additional Editor Comments (optional):

Thank you for considering the comments raised and addressing them adequately

Reviewers' comments:

Reviewer's Responses to Questions

**Comments to the Author**

1. If the authors have adequately addressed your comments raised in a previous round of review and you feel that this manuscript is now acceptable for publication, you may indicate that here to bypass the “Comments to the Author” section, enter your conflict of interest statement in the “Confidential to Editor” section, and submit your "Accept" recommendation.

Reviewer #1: All comments have been addressed

Reviewer #2: All comments have been addressed

2. Is the manuscript technically sound, and do the data support the conclusions?

Reviewer #1: Yes

Reviewer #2: Yes

3. Has the statistical analysis been performed appropriately and rigorously? 

Reviewer #1: Yes

Reviewer #2: Yes

4. Have the authors made all data underlying the findings in their manuscript fully available?

Reviewer #1: No

Reviewer #2: Yes

5. Is the manuscript presented in an intelligible fashion and written in standard English?

Reviewer #1: Yes

Reviewer #2: Yes

6. Review Comments to the Author

Reviewer #1: Thank you for the opportunity to review this manuscript which evaluates the reliability of billing code assignment by a pediatric emergency physician compared to that assigned by a billing auditor. The revision largely attends to the reviewers comments and the conclusions are supported by the data presented. The change in title better reflects what was done in this study.

Reviewer #2: (No Response)

7. PLOS authors have the option to publish the peer review history of their article (what does this mean?). If published, this will include your full peer review and any attached files.

Reviewer #1: **Yes: **Rafiuk Cosmos Yakubu

Reviewer #2: No

---

## [Editor Report · Acceptance letter]

18 Aug 2023

PONE-D-23-06963R1 

Assessing the reliability of pediatric emergency medicine billing code assignment for future consideration as a proxy workload measure 

Dear Dr. Park:

I'm pleased to inform you that your manuscript has been deemed suitable for publication in PLOS ONE. Congratulations! Your manuscript is now with our production department. 

Kind regards, 

on behalf of

Dr. Benjamin Demah Nuertey 

Academic Editor

PLOS ONE